# Learning to Generate 3D Shapes with Generative Cellular Automata

**Dongsu Zhang, Changwoon Choi, Jeonghwan Kim & Young Min Kim**
Department of Electrical and Computer Engineering, Seoul National University
`{96lives, zzzmaster, whitealex95, youngmin.kim}@snu.ac.kr`

## Abstract

We present a probabilistic 3D generative model, named Generative Cellular Automata, which is able to produce diverse and high quality shapes. We formulate the shape generation process as sampling from the transition kernel of a Markov chain, where the sampling chain eventually evolves to the full shape of the learned distribution. The transition kernel employs the local update rules of cellular automata, effectively reducing the search space in a high-resolution 3D grid space by exploiting the connectivity and sparsity of 3D shapes. Our progressive generation only focuses on the sparse set of occupied voxels and their neighborhood, thus enabling the utilization of an expressive sparse convolutional network. We propose an effective training scheme to obtain the local homogeneous rule of generative cellular automata with sequences that are slightly different from the sampling chain but converge to the full shapes in the training data. Extensive experiments on probabilistic shape completion and shape generation demonstrate that our method achieves competitive performance against recent methods.

## 1 Introduction

Probabilistic 3D shape generation aims to learn and sample from the distribution of diverse 3D shapes and has applications including 3D contents generation or robot interaction. Specifically, learning the distribution of shapes or scenes can automate the process of generating diverse and realistic virtual environments or new object designs. Likewise, modeling the conditional distribution of the whole scene given partial raw 3D scans can help the decision process of a robot, by informing various possible outputs of occluded space.

The distribution of plausible shapes in 3D space is diverse and complex, and we seek a scalable formulation of the shape generation process. Pioneering works on 3D shape generation try to regress the entire shape (Dai et al. (2017)) which often fail to recover fine details. We propose a more modular approach that progressively generates shape by a sequence of local updates. Our work takes inspiration from prior works on autoregressive models in the image domains, such as the variants of pixelCNN (van den Oord & Kalchbrenner (2016); van den Oord et al. (2016; 2017)), which have been successful in image generation. The key idea of pixelCNN (van den Oord et al. (2016)) is to order the pixels, and then learn the conditional distribution of the next pixel given all of the previous pixels. Thus generating an image becomes the task of sampling pixel-by-pixel in the predefined order. Recently, PointGrow (Sun et al. (2020)) proposes a similar approach in the field of 3D generation, replacing the RGB values of pixels with the coordinates of points and sampling point-by-point in a sequential manner. While the work proposes a promising interpretable generation process by sequentially growing a shape, the required number of sampling procedures expands linearly with the number of points, making the model hard to scale to high-resolution data.

We believe that a more scalable solution in 3D is to employ the local update rules of cellular automata (CA). CA, a mathematical model operating on a grid, defines a state to be a collection of cells that carries values in the grid (Wolfram (1982)). The CA repeatedly mutates its states based on the predefined homogeneous update rules only determined by the spatial neighborhood of the current cell. In contrast to the conventional CA where the rules are predefined, we employ a neural network to infer the stochastic sequential transition rule of individual cells based on Markov chain. The obtained homogeneous local rule for the individual cells constitutes the 3D generative model, named

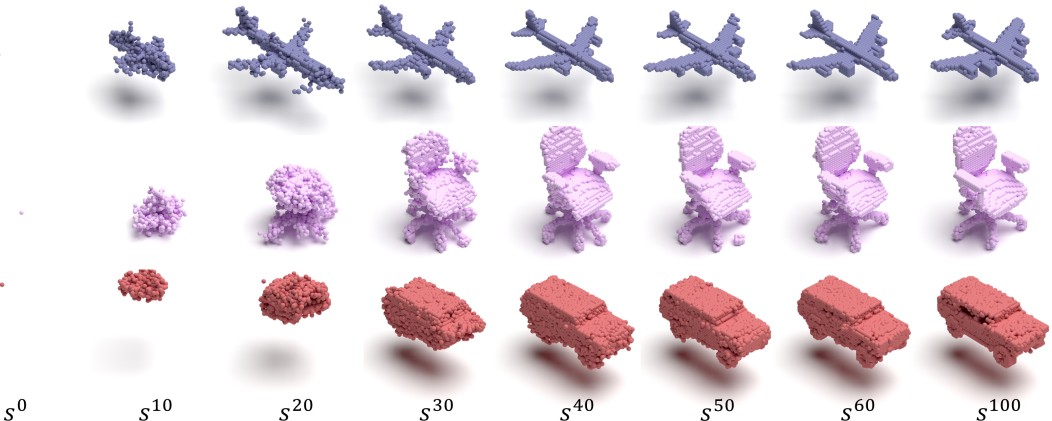

$s^0$     $s^{10}$     $s^{20}$     $s^{30}$     $s^{40}$     $s^{50}$     $s^{60}$     $s^{100}$

Figure 1: Sampling chain of shape generation. Full generation processes are included in the supplementary material.

Generative Cellular Automata (GCA). When the rule is distributed into the group of occupied cells of an arbitrary starting shape, the sequence of local transitions eventually evolves into an instance among the diverse shapes from the multi-modal distribution. The local update rules of CA greatly reduce the search space of voxel occupancy, exploiting the sparsity and connectivity of 3D shapes.

We suggest a simple, progressive training procedure to learn the distribution of local transitions of which repeated application generates the shape of the data distribution. We represent the shape in terms of surface points and store it within a 3D grid, and the transition rule is trained only on the occupied cells by employing a sparse CNN (Graham et al. (2018)). The sparse representation can capture the high-resolution context information, and yet learn the effective rule enjoying the expressive power of deep CNN as demonstrated in various computer vision tasks (Krizhevsky et al. (2012); He et al. (2017)). Inspired by Bordes et al. (2017), our model learns sequences that are slightly different from the sampling chain but converge to the full shapes in the training data. The network successfully learns the update rules of CA, such that a single inference samples from the distribution of diverse modes along the surface.

The contributions of the paper are highlighted as follows: (1) We propose Generative Cellular Automata (GCA), a Markov chain based 3D generative model that iteratively mends the shape to a learned distribution, generating diverse and high-fidelity shapes. (2) Our work is the first to learn the local update rules of cellular automata for 3D shape generation in voxel representation. This enables the use of an expressive sparse CNN and reduces the search space of voxel occupancy by fully exploiting sparsity and connectivity of 3D shapes. (3) Extensive experiments show that our method has competitive performance against the state-of-the-art models in probabilistic shape completion and shape generation.

## 2   3D Shape Generation with Generative Cellular Automata

Let $\mathbb{Z}^n$ be an $n$-dimensional uniform grid space, where $n = 3$ for a 3D voxel space. A 3D shape is represented as a state $s \subset \mathbb{Z}^3$, which is an ordered set of occupied cells $c \in \mathbb{Z}^3$ in a binary occupancy grid based on the location of the surface. Note that our voxel representation is different from the conventional occupancy grid, where 1 represents that the cell is inside the surface. Instead, we only store the cells lying on the surface. This representation can better exploit the sparsity of 3D shape than the full occupancy grid.

The shape generation process is presented as a sequence of state variables $s^{0:T}$ that is drawn from the following Markov Chain:

$$s^0 \sim p^0 \quad s^{t+1} \sim p_\theta(s^{t+1}|s^t) \tag{1}$$

where $p^0$ is the initial distribution and $p_\theta$ is the homogeneous transition kernel parameterized by $\theta$. We denote the sampled sequence $s^0 \to s^1 \to \cdots \to s^T$ as a *sampling chain*. Given the data set $\mathcal{D}$ containing 3D shapes $x \in \mathcal{D}$, our objective is to learn the parameters $\theta$ of transition kernel $p_\theta$, such that the marginal distribution of final generated sample $p(s^T) = \sum_{s^{0:T-1}} p^0(s^0) \prod_{0 \le t < T} p_\theta(s^{t+1}|s^t)$

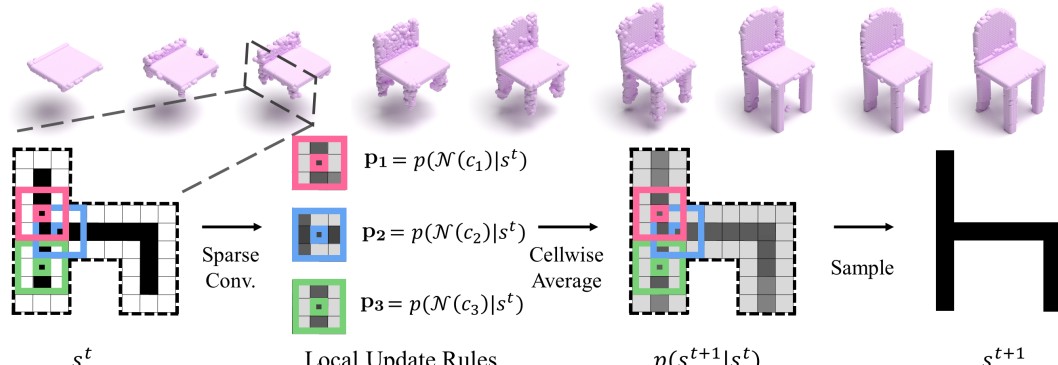

Figure 2: Overview of GCA. The top shows a probabilistic shape completion sampling chain of a chair starting from the partial input. Bottom figures show an illustrative figure of a single transition. Starting from state $s^t$ of occupied voxels, we apply sparse CNN to obtain the neighborhood occupancy probability. Then we aggregate the probabilities by cell-wise averaging and sample the next state.

is close to the data distribution. The initial state $s^0$ can be defined differently depending on the task to solve. For the task of probabilistic shape completion, $s^0$ is given as the partial input shape. For shape generation, we set the initial state $s^0$ to be the most simple state we can think of, a single cell $\{c\}$. Figure 1 presents examples of sampling chains of shape generation, where the starting shape $s^0$ is merely a single cell.

The GCA further splits the transition kernel $p_\theta(s^{t+1}|s^t)$ to be the combination of local update rules on individual occupied cells $c_i \in s^t$, as depicted in Figure 2. The cellular transition is implemented with the sparse convolution, which is translation invariant if implemented with a fully convolutional network, and outputs the distribution of local occupied cells. Then individual predictions are aggregated by cell-wise averaging, resulting in the binary probability distribution for occupancy of each cell that follows the Bernoulli distribution:

$$p_\theta(s^{t+1}|s^t) = \prod_{c \in \mathbb{Z}^n} p_\theta(c|s^t). \tag{2}$$

The next state $s^{t+1}$ is sampled independently for individual cells from the obtained distribution and the sampling chain continues to the next time step.

For each transition $p_\theta(s^{t+1}|s^t)$, we limit the search space of the occupied cells by confining our predictions within the neighborhood of the occupied cells. The underlying assumption is that the occupied cells of a valid shape are connected and a successful generation is possible by progressive growing into the immediate neighborhood of the given state. Specifically, the output of the sparse convolution is the occupancy probability of neighborhood cells $p_i = p_\theta(\mathcal{N}(c_i)|s^t)$, where the neighborhood cells are those that fall within a radius $r$ ball centered at the cell, $\mathcal{N}(c_i) = \{c' \in \mathbb{Z}^n | d(c_i, c') \le r\}$ given a distance metric $d$. Other cells are ignored assuming they have probability 0 of occupancy. If the input state has $M$ occupied cells $s^t = \{c_1, \cdots, c_M\}$, the sparse convolution predicts the occupancy probability of individual cells with $N$-dimension vectors $\mathcal{P} = \{p_1, \cdots, p_M\}$, where $N$ is the number of neighborhood cells fixed by the distance threshold $r$ within the uniform grid $\mathbb{Z}^n$. After the cell-wise averaging step, the aggregated probability is nonzero for coordinates in $\mathcal{N}(s^t) = \bigcup_{c \in s^t} \mathcal{N}(c)$. Then the cell-wise sampling in Eq. (2) is performed only within $\mathcal{N}(s^t)$, instead of the full grid $\mathbb{Z}^n$, leading to an efficient sampling procedure.

The stochastic local transition rule $p_\theta(\mathcal{N}(c_i)|s^t)$ changes the state of a cell's immediate neighborhood $N(c_i)$, but the inference is determined from a larger perception neighborhood. In contrast, classical cellular automata updates a state of a cell determined by a fixed rule given the observation of the cell's immediate neighborhood. The large perception neighborhood of GCA is effectively handled by deep sparse convolutional network, and results in convergence to a single consistent global shape out of diverse possible output shapes as further discussed in the appendix F.

## 3 TRAINING GENERATIVE CELLULAR AUTOMATA

We aim to learn the parameters for the local transition probability $p_\theta(\mathcal{N}(c_i)|s^t)$, whose repetitive application generates shape that follows the complex distribution of the data set. If we have sequences of sampling chains, then all current and next states can serve as the training data. Because we only have the set of complete shapes $\mathcal{D}$, we emulate the sequence of sampling chain and obtain the intermediate states.

The emulated sequence of a sampling chain may start from an arbitrary state, and needs to converge to the full shape $x \in \mathcal{D}$ after local transitions to the neighbor of the previous state. A naive method would be to sample the next state $s^t$ from the sampling chain and maximize $p_\theta(x \cap \mathcal{N}(s^t)|s^t)$, the probability of the shape that is closest to $x$ among reachable forms from the current state[1], similar to scheduled sampling (Bengio et al. (2015)). While this approach clearly emulates the inference procedure, it leads to learning a biased estimator as pointed out in Huszár (2015). More importantly, the emulated sequence cannot consistently learn the full shape $x$ as it is not guaranteed to visit the state $s$ such that $x \subset \mathcal{N}(s)$.

We instead employ the use of infusion training procedure suggested by Bordes et al. (2017). Specifically, the *infusion chain*, denoted as $\tilde{s}^0 \to \tilde{s}^1 \to ... \to \tilde{s}^T$, is obtained as following:

$$\tilde{s}^0 \sim q^0(\tilde{s}^0|x) \quad q^t(\tilde{s}^{t+1}|\tilde{s}^t, x) = \prod_{\tilde{c} \in \mathcal{N}(\tilde{s}^t)} (1 - \alpha^t)p_\theta(\tilde{c}|\tilde{s}^t) + \alpha^t \delta_x(\tilde{c}) \tag{3}$$

where $q^0$ indicates the initial distribution of infusion chain, and $q^t$ is the infusion transition kernel at time step $t$. For probabilistic shape completion $\tilde{s}^0$ is sampled as a subset of $x$, while for shape generation $\tilde{s}^0$ is a single cell $c \in x$. The transition kernel $q^t$ is defined for cells in the neighborhood $\tilde{c} \in \mathcal{N}(\tilde{s}_t)$ as the mixture of $p_\theta(\tilde{c}|\tilde{s}^t)$ and $\delta_x(\tilde{c})$, which are the transition kernel of the sampling chain and the infusion of the ground shape $x$, respectively. $\delta_x(\tilde{c})$ is crucial to guarantee the sequence to ultimately reach the ground truth shape, and is formulated as the Bernoulli distribution with probability 1, if $\tilde{c} \in x$, else 0. We set the infusion rate to increase linearly with respect to time step, i.e., $\alpha^t = \max(wt, 1)$, where $w > 0$ is the speed of infusion rate as in Bordes et al. (2017).

We can prove that the training procedure converges to the shape $x$ in the training data if it is composed of weakly connected cells. We first define the connectivity of two states.

**Definition 1.** *We call a state $\tilde{s}'$ to be **partially connected** to state $x$, if for any cell $b \in x$, there is a cell $c_0 \in \tilde{s}' \cap x$, and a finite sequence of coordinates $c_{0:T_b}$ in $x$ that starts with $c_0$ and ends with $c_{T_b} = b$, where each subsequent element is closer than the given threshold distance $r$, i.e., for any $b \in x, \exists c_{0:T_b}$, such that $c_i \in x, d(c_i, c_{i+1}) \leq r, 0 \leq i \leq T_b$ and $c_0 \in \tilde{s}', c_{T_b} = b$.*

The shape $x$ is partially connected to any $\tilde{s}' \neq \emptyset$ if we can create a sequence of coordinates between any pair of cells in $x$ that is composed of local transitions bounded by the distance $r$. This is a very weak connectivity condition, and any set that overlaps with $x$ is partially connected to $x$ for shapes with continuous surfaces, which include shapes in the conventional dataset.

Now assuming that the state $\tilde{s}^{t'}$ is partially connected to the state $x$, we recursively create a sequence by defining $\tilde{s}^{t+1} = \mathcal{N}(\tilde{s}^t) \cap x$, which is the next sequence of infusion chain with infusion rate 1. Since we use a linear scheduler for infusion rate, we can assume that there exists a state $\tilde{s}^{t'}$ such that infusion rate $\alpha^{t'} = 1$. The following proposition proves that the sequence $(\tilde{s}^t)_{t \geq t'}$ converges to $x$ with local transitions.

**Proposition 1.** *Let state $\tilde{s}^{t'}$ be partially connected to state $x$, where $x$ has a finite number of occupied cells. We denote a sequence of states $\tilde{s}^{t':\infty}$, recursively defined as $\tilde{s}^{t+1} = \mathcal{N}(\tilde{s}^t) \cap x$. Then, there exists an integer $T'$ such that $\tilde{s}^t = x, t \geq T'$.*

*Proof.* The proof is found in the appendix A. $\qquad\square$

The proposition states that the samples of infusion chain eventually converge to the shape $x$, and thus we can compute the nonzero likelihood $p(x|\tilde{s}^T)$ during training if $T$ is large enough. One can also interpret the training procedure as learning the sequence of states that converges to $x$ and is close to

---

[1]Since we defined a state as a set of occupied cells, union ($\cup$), intersection ($\cap$), and subset ($\subset$) operations can be defined as regular sets.

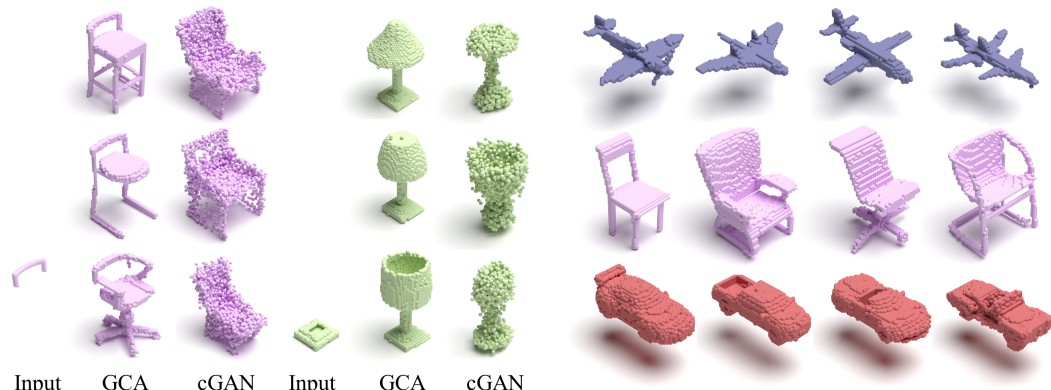

Figure 3: Qualitative comparison of probabilistic shape completion.

Figure 4: Samples from shape generation.

the sampling chain. We empirically observe that most of the training samples converge to the shape $x$ before the infusion rate becomes 1.

The training procedure can be summarized as the following:

1. Sample infusion chain $\tilde{s}^{0:T}$ from $\tilde{s}^0 \sim q^0(\tilde{s}^0|x)$, $\tilde{s}^{t+1} \sim q^t(\tilde{s}^{t+1}|\tilde{s}^t, x)$.
2. For each state $\tilde{s}^t$, maximize the log-likelihood that has the closest form to $x$ via gradient descent, i.e., $\theta \leftarrow \theta + \eta \frac{\partial \log p_\theta(x \cap \mathcal{N}(\tilde{s}^t)|\tilde{s}^t)}{\partial \theta}$.

The full training algorithm utilizes a data buffer to de-correlate the gradients of subsequent time steps, and accelerates the process by controlling the time steps based on the current state. More details regarding the full training algorithm can be found in the appendix B.

# 4 EXPERIMENTS

We demonstrate the ability of GCA to generate high-fidelity shapes in two tasks: probabilistic shape completion (Sec. 4.1) and shape generation (Sec. 4.2).

State-of-the-art works on both tasks generate shapes using point cloud whereas ours uses grid structure. For comparison, we extract high-resolution points from the CAD model and map them into $64^3$ voxel grid. Individual shapes in the dataset are centered and uniformly scaled to fit within the cube $[-1, 1]^3$. We also present additional analysis on how the sparse representation and local transition of GCA can successfully learn to generate fine resolution shapes with continuous geometry (Sec. 4.3). Details regarding the training settings and baselines are reported in the appendix G.

## 4.1 PROBABILISTIC SHAPE COMPLETION

**Dataset and implementation details.** The probabilistic shape completion is tested with PartNet (Mo et al. (2019)) and PartNet-Scan (Wu et al. (2020)) dataset, where objects in ShapeNet(Chang et al. (2015)) are annotated with instance-level parts. For each instance, we identify the input partial object by randomly choosing parts from the complete shape, and generating a diverse set of complete shapes. PartNet-Scan dataset simulates the real-world situation where partial scans suffer from part-level incompleteness by randomly selecting parts from the PartNet and virtually scanning the remaining parts. We validate our approach on chair, lamp, and table categories following Wu et al. (2020). We use neighborhood radius $r = 3$, $T = 70$ with infusion speed $w = 0.005$ for all datasets. Following the work of Wu et al. (2020), for each partial shape in the test set, we generate ten completion results and sample 2,048 points from occupied cells.

We compare the performance of probabilistic shape completion of GCA against cGAN (Wu et al. (2020)), cGAN and its variations (cGAN-im-l2z and cGAN-im-pc2z) [2], with 3D-IWGAN (Smith

---

[2]We would like to acknowledge that, when training, cGAN only requires the partial set and complete shape set without any explicit pairing between each of the instances in the set. However, we report cGAN as the baseline

| PartNet | MMD (quality, ↓) | | | | TMD (diversity, ↑) | | | | UHD (fidelity, ↓) | | | |
|---|---|---|---|---|---|---|---|---|---|---|---|---|
| Method | Chair | Lamp | Table | Avg. | Chair | Lamp | Table | Avg. | Chair | Lamp | Table | Avg. |
| 3D-IWGAN | 1.94 | 3.57 | 8.18 | 4.56 | 0.99 | 3.58 | 1.52 | 2.03 | 6.89 | 8.86 | 8.07 | 7.94 |
| cGAN-im-l2z | 1.74 | 2.36 | 1.68 | 1.93 | 3.74 | 2.68 | 3.59 | 3.34 | 8.41 | 6.37 | 7.21 | 7.33 |
| cGAN-im-pc2z | 1.90 | 2.55 | 1.54 | 2.00 | 1.01 | 0.56 | 0.51 | 0.69 | 6.65 | **5.40** | **5.38** | **5.81** |
| cGAN | 1.52 | 1.97 | 1.46 | 1.65 | 2.75 | 3.31 | 3.30 | 3.12 | 6.89 | 5.72 | 5.56 | 6.06 |
| GCA | **1.28** | **1.85** | **1.13** | **1.42** | **4.74** | **9.38** | **4.50** | **6.20** | **6.21** | 5.96 | 5.60 | 5.92 |
| GCA (joint) | 1.33 | 1.89 | 1.14 | 1.45 | 3.69 | 13.49 | 4.55 | 7.24 | 6.14 | 6.31 | 5.59 | 6.01 |

| PartNet-Scan | MMD (quality, ↓) | | | | TMD (diversity, ↑) | | | | UHD (fidelity, ↓) | | | |
|---|---|---|---|---|---|---|---|---|---|---|---|---|
| Method | Chair | Lamp | Table | Avg. | Chair | Lamp | Table | Avg. | Chair | Lamp | Table | Avg. |
| cGAN-im-l2z | 1.79 | 2.58 | 1.92 | 2.10 | **3.85** | 3.18 | **4.75** | 3.93 | 7.88 | 6.39 | 7.40 | 7.22 |
| cGAN-im-pc2z | 1.65 | 2.75 | 1.84 | 2.08 | 1.91 | 0.50 | 1.86 | 1.42 | 7.50 | **5.36** | 5.68 | 6.18 |
| cGAN | **1.53** | 2.15 | 1.58 | 1.75 | 2.91 | 4.16 | 3.88 | 3.65 | 6.93 | 5.74 | 6.24 | 6.30 |
| GCA | 1.55 | **1.97** | **1.57** | **1.70** | 1.73 | **10.61** | 2.89 | **5.20** | **6.15** | 5.86 | **5.54** | **5.85** |

Table 1: Quantitative comparison of probabilistic shape completion results on PartNet (top) and PartNet-Scan (bottom). The best results trained in single class are in bold. Note that MMD (quality), TMD (diversity) and UHD (fidelity) presented in the tables are multiplied by $10^3$, $10^2$ and $10^2$, respectively. GCA (joint) denotes GCA trained on all categories combined, but evaluated individually.

& Meger (2017)) The qualitative results in Figure 3 and the appendix H clearly show that our high-resolution completion exhibits sophisticated geometric details compared to the state-of-the-art. Our superior performance is further verified in the quantitative results reported in Table 1 with three metrics: minimal matching distance (MMD), total mutual difference (TMD), and unidirectional Hausdorff distance (UHD). MMD and TMD are reported after generated samples are centered due to translation invariant nature of our model. The detailed description regarding the evaluation metrics are found in the appendix G.3.

We outperform all other methods in MMD and TMD on average, and achieve comparable or the best results in UHD. The results indicate that we can generate high fidelity (MMD) yet diverse (TMD) shapes while being loyal to the given partial geometry (UHD). We observe an extremely large variation of completion modes in the lamp dataset, which is the dataset with the most diverse structure while having a high level of incompleteness in the input partial data. This is because the lamp dataset contains fragmented small parts that could be selected as an highly incomplete input. We include further discussion about the diversity in the lamp dataset in the appendix E.

**Multiple category completion**. While previous works on shape completion are trained and tested individual categories separately, we also provide the results on training with the entire category, denoted as GCA (joint) in Table 1. The performance of jointly trained transition kernel does not significantly differ to those of independently trained, demonstrating the expressiveness and high capacity of GCA. In addition, interesting behavior arises with the jointly trained GCA. As shown in the left image of Figure 5, given ambiguous input shape, the completed shape can sometimes exhibit features from categories other than that of the given shape.

Furthermore, GCA is capable of completing a scene composed of multiple objects although it is trained to complete a single object (Figure 5, right). We believe that this is due to the translation invariance and the local transition of GCA. As discussed in Section 2, the deep network observes relatively large neighborhood, but GCA updates the local neighborhood based on multiple layers of sparse convolution which is less affected by distant voxels. The information propagation between layers of sparse convolutional network (Graham et al. (2018)) is mediated by occupied voxels, which effectively constrains the perception neighborhood to the connected voxels. As a result, the effect of separated objects are successfully controlled. We also include additional examples in the appendix D on how the GCA trained in one category completes the unseen category.

## 4.2 SHAPE GENERATION

**Dataset and implementation details.** We test the performance of shape generation using the ShapeNet (Chang et al. (2015)) dataset, focusing on the categories with the most number of shapes:

---

to our method, since this is the most recent probabilistic shape completion. cGAN-im-l2z and cGAN-im-pc2z are variants of cGAN that implicitly model the multi-modality for probabilistic shape completion.

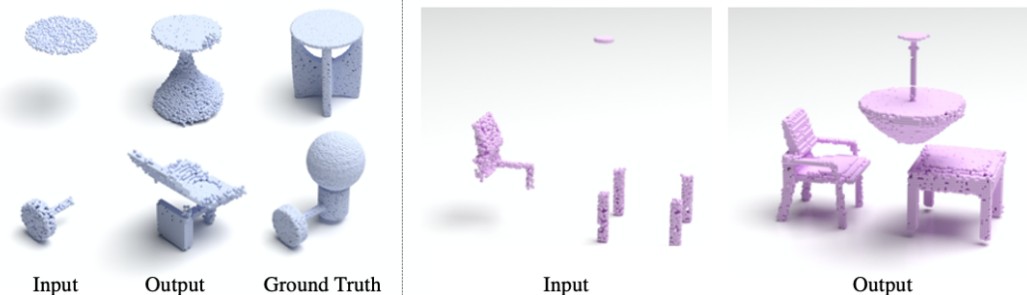

Figure 5: Shape completion results when jointly trained with multiple categories of shapes. Given an ambiguous input shape, features from different categories co-exist (top: lamp + table, bottom: lamp + chair) in the completed shape (left). Even though GCA is trained to complete a *single* object, it can complete a scene with multiple partial shapes of different categories (right).

| ShapeNet | 1-NNA (distance of distributions, ↓) | | | | COV (diversity, ↑) | | | | MMD (quality, ↓) | | | |
|---|---|---|---|---|---|---|---|---|---|---|---|---|
| Method | Airplane | Car | Chair | Avg. | Airplane | Car | Chair | Avg. | Airplane | Car | Chair | Avg. |
| r-GAN | 95.80 | 99.29 | 84.82 | 93.30 | 38.52 | 8.24 | 19.49 | 22.08 | 1.66 | 6.23 | 18.19 | 8.69 |
| GCN | 95.16 | - | 86.52 | - | 9.38 | - | 6.95 | - | 2.62 | - | 23.10 | - |
| Tree-GAN | 95.06 | - | 74.55 | - | 44.69 | - | 40.33 | - | 1.47 | - | 16.15 | - |
| 3D-IWGAN | 94.70 | 76.60 | **63.62** | 78.31 | 38.02 | 42.90 | 45.17 | 42.03 | 1.53 | 4.33 | 15.71 | 7.19 |
| PointFlow | **83.21** | 68.75 | 67.60 | 73.19 | 39.51 | 39.20 | 40.94 | 39.88 | 1.41 | 4.21 | 15.03 | 6.88 |
| ShapeGF | 85.06 | 65.48 | 66.16 | **72.23** | **47.65** | 44.60 | **46.37** | **46.21** | 1.29 | **4.09** | **14.82** | **6.73** |
| GCA | 90.62 | **65.06** | 65.71 | 74.85 | 35.80 | **46.31** | 44.56 | 40.79 | **1.25** | 4.19 | 16.89 | 7.44 |
| Train set | 72.10 | 52.98 | 53.93 | 59.67 | 45.43 | 48.30 | 50.45 | 48.06 | 1.29 | 4.21 | 15.89 | 7.13 |

Table 2: Quantitative comparison results of shape generation on ShapeNet. The best results are in bold. Note that MMD is multiplied by $10^3$.

airplane, car, and chair, as presented in Yang et al. (2019). We use the same experimental setup as Cai et al. (2020), and sample 2,048 points from the occupied cells of the generate shape and center the shape, due to translation invariant aspect of our model, for a fair comparison. We use neighborhood size $r = 2$ with $L_1$ distance metric, $T = 100$ inferences, infusion speed $w = 0.005$ for airplane and car dataset, and $r = 3$ for chair category, which tend to converge better.

The quantitative results of shape generation are reported in Table 2. We compare the performance of our approach against recent point cloud based methods: r-GAN (Achlioptas et al. (2018)), GCN-GAN (Valsesia et al. (2019)), Tree-GAN (Shu et al. (2019)), Pointflow (Yang et al. (2019)), ShapeGF (Cai et al. (2020)), a voxel-based method: 3D-IWGAN (Smith & Meger (2017)), and training set. The scores assigned to training set can be regarded as an optimal value for each metric. We evaluate our model on three metrics, 1-nearest-neighbor-accuracy (1-NNA), coverage (COV), and minimal matching distance (MMD) as defined in the appendix G.3.

Our approach achieves the state-of-the-art results on 1-NNA of car category. As remarked by Yang et al. (2019), 1-NNA is superior on measuring the distributional similarity between generated set and test set from the perspective of both diversity and quality. Our method also achieves state-of-the-art results on COV of car, implying that our method is able to generate diverse results. Note we are using a homogeneous transition kernel confined within the local neighborhood of occupied cells, but can still achieve noticeable performance on generating a rich distribution of shape. MMD is a conventional metric to represent the shape fidelity, but some results achieve scores better than the training set which might be questionable (Yang et al. (2019)). We visualize our shape generation results in Figure 4 and in the appendix I.

### 4.3 ANALYSIS ON NEIGHBORHOOD AND CONNECTIVITY

GCA makes high-resolution predictions by only handling the occupied voxels and their neighborhood, thus overcoming the limitation of memory requirement of voxel-based representation of 3D shape. In Figure 6, we empirically analyze the required search space during the shape generation process that starts from a single cell and evolves to the full shape. The occupied voxels take approximately 2% of

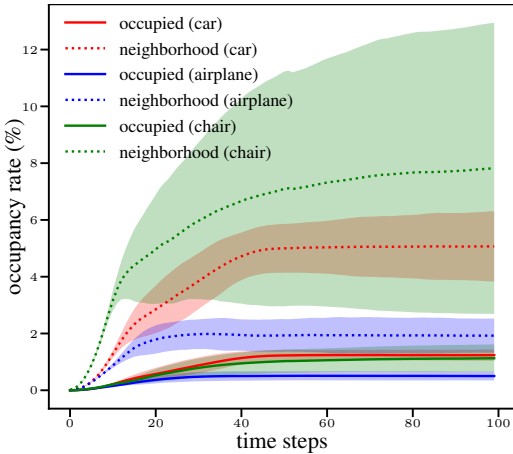

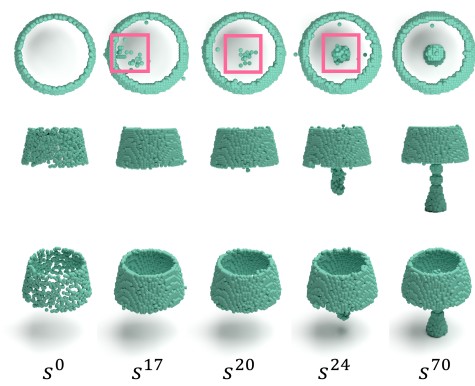

$s^0 \quad s^{17} \quad s^{20} \quad s^{24} \quad s^{70}$

Figure 6: Search space visualization. The graph shows the percentage of the occupied voxels (solid lines) and the searched neighborhood voxels (dashed lines) during 100 time steps of shape generation process.

Figure 7: Completing disconnected shapes. Top, front, and side views of generation process are shown with $r = 3$, $L_1$ metric neighborhood, where the diameter of the lampshade is over 30. Given the partial shape of the shade, GCA is able to generate the body of the lamp that is disconnected from the input.

the total volume (solid lines), and their neighborhood adds up to 2-8% (dashed lines). After a certain amount of time steps, the growing speed of the search space gradually decreases, implying that our method tends to converge and maintain the generated shape.

The large structural variation observed in the lamp dataset leads to interesting observation on the neighborhood size of GCA and the connectivity of the shape. When $r = 1$, nearly 10% of the infusion chain data fails to meet the stopping condition (cover more than 95% of $x$, see the training detail in the appendix B) with time step $T = 100$. This is because GCA trained on a small neighborhood size, while enabling the transition kernel to be trained quickly, lacks the flexibility to model disconnected shapes even after large time steps. On the other hand, increased neighborhood size can capture disjoint parts at the expense of larger search space and instability. When $r = 10$, the trained transition kernel generates output that is noisy or unrelated to the input partial shape. We believe that the transition kernel can not be trained to cover the entire search space as the neighborhood size increases cubic to $r$. A few bad sampling steps can lead to a state that is far from the previous state or unobserved during training. The effects of hyperparameters are further discussed in the appendix C.

We also clarify the notion of connectivity in relation to the neighborhood size. While we mostly demonstrate the performance of GCA generating 3D shapes with continuous surface, GCA can surely generate disjoint parts as long as the distance between parts is within the radius $r$. Besides, GCA is flexible enough to learn to generate parts that are farther than $r$. $r$ is merely the distance of cells between adjacent steps, and it is possible to learn the sequence of states that generates a temporary *bridge* to reach disconnected parts after a few steps. Figure 7 shows an observed sequence of a sampling chain that generates a temporary bridge from the shade to the body of the lamp further than the neighborhood size. After the sampling chain reaches the main body, GCA removes the bridge and successfully generates the full shape.

# 5 RELATED WORKS

## 5.1 GENERATIVE MODELS

**Autoregressive models.** PixelCNN and its variants (van den Oord & Kalchbrenner (2016); van den Oord et al. (2016)) sequentially sample a single pixel value at a time, conditioned on the previously generated pixels. PointGrow (Sun et al. (2020)) extends the idea to 3D by sampling the coordinates of points in a sequential manner. While these approaches lead to tractable probabilistic density estimation, inferring one pixel/point with a single inference is not scalable to high resolution 3D voxel space. GCA, on the other hand, can grow to the shape occupying any number of voxels in the neighborhood at each inference.

**Markov chain models.** Previous works learn the transition kernel of Markov chain operating on the input space which eventually produces samples matching the data distribution (Sohl-Dickstein et al. (2015); Anirudh et al. (2017); Bordes et al. (2017)). The diffusion-based models (Sohl-Dickstein et al. (2015); Anirudh et al. (2017)) start from a random noise distribution and incrementally denoise the input to generate the data. They learn the reverse of a diffusion process, which begins from the data and gradually adds noise to become a factorial noise distribution. On the other hand, the work of infusion training (Bordes et al. (2017)) slightly infuses data to input dimension during training, and learns the chain biased towards the data. This removes the process of inverting the diffusion chain and allows the chain to start from *any* distribution. The training of GCA utilizes the technique of infusion, enabling to train on both shape generation and completion.

**3D generative models**. Recent works on 3D generative models (Yang et al. (2019); Cai et al. (2020); Wu et al. (2020)) achieve noticeable success with point cloud based representation. They are based on PointNet (Qi et al. (2016)), one of the most widely-used neural network architecture consuming point cloud. PointNet condenses the 3D shape into a single global feature vector to achieve permutation invariance, but in turn lacks the representation power of local context observed in deep neural architecture. On the other hand, voxel-based generation methods (Wu et al. (2016); Smith & Meger (2017)) can capture local features by directly adapting a CNN-based architecture, but suffer from immense memory consumption in high-resolution. Recent works of sparse CNN (Graham et al. (2018); Choy et al. (2019)) solves the problem of voxel representation by exploiting sparsity, and outperforms other methods by a significant margin in the field of 3D semantic segmentation. To the best of our knowledge, we are the first to utilize the expressive power of sparse CNN to learn a probabilistic 3D generative model in a high-resolution 3D grid.

## 5.2 CELLULAR AUTOMATA

Cellular automata (CA) is introduced to simulate biological or chemical systems (Wolfram (1982); Markus & Hess (1990)). CA consists of a grid of cells where individual cells are repeatedly updated with the rule depending only on the state of neighborhood cells. Each cell is updated with the same set of local rules, but when the update is repeatedly distributed in the entire grid, complex and interesting behavior can emerge. However, CA requires a fixed rule to be given. Wulff & Hertz (1992) learn simple rules with a neural network and model the underlying dynamics of CA. Neural Cellular Automata (Mordvintsev et al. (2020)) shows an interesting extension of CA that incorporates neural network to learn the iterative update rule and successfully generates a pre-specified image. While the benefits of CA are not clear in the image domain, we show that the local update rule of CA can excel in generating 3D shape, which is sparse and connected, by achieving near state-of-the-art performance.

There is one major difference of GCA compared to the classical CA. The transition kernel employs the deep sparse CNN and the effective perception neighborhood is much larger than the update neighborhood. With the extension, GCA has the capacity to perceive the complex context and concurrently generates high-fidelity local shape as demonstrated by multi-category and multi-object generation in Figure 5.

## 6 CONCLUSION AND FUTURE WORK

In this work, we present a probabilistic 3D generative model, named GCA, capable of producing diverse and high-fidelity shapes. Our model learns a transition kernel of a Markov chain operating only on the spatial vicinity of an input voxel shape, which is the local update rules of cellular automata. We effectively reduce the search space in a high resolution voxel space, with the advantage of employing a highly expressive sparse convolution neural network that leads to state-of-the-art performance in probabilistic completion and yields competitive results against recent methods in shape generation.

Although our experiments are mainly focused on shape generation, we believe our framework is capable of modeling a wide range of sequential data, which tends to be sparse and connected. For instance, particle-based fluid simulation models continuous movement of fluid particles, where the local update rules of our approach can be applied.

ACKNOWLEDGMENTS

We thank Youngjae Lee for helpful discussion and advice. This research was supported by the National Research Foundation of Korea (NRF) grant funded by the Korea government (MSIT) (No. 2020R1C1C1008195) and the National Convergence Research of Scientific Challenges through the National Research Foundation of Korea (NRF) funded by Ministry of Science and ICT (NRF-2020M3F7A1094300).

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

## A    PROOF OF PROPOSITION 1

We present the proposition in Sec. 3 and its proof.

**Proposition 1.** *Let state $\tilde{s}^{t'}$ be partially connected to state $x$, where $x$ has a finite number of occupied cells. We denote a sequence of states $\tilde{s}^{t':\infty}$, recursively defined as $\tilde{s}^{t+1} = \mathcal{N}(\tilde{s}^t) \cap x$. Then, there exists an integer $T'$ such that $\tilde{s}^t = x$, $t \geq T'$.*

*Proof.* If $\tilde{s}^{T'} = x$, $\tilde{s}^{T'+1} = \mathcal{N}(\tilde{s}^{T'}) \cap x = x$ and $\tilde{s}^t = x$ for $t \geq T'$. Now we show that there exists an integer $T'$ such that $\tilde{s}^{T'} = x$ by showing $\tilde{s}^{T'} \subset x$ and $\tilde{s}^{T'} \supset x$. We first prove the former by showing that sequence of states $\tilde{s}^{t':\infty}$ is an increasing sequence bounded by $x$. $\forall t > t'$,

$$\tilde{s}^t = \tilde{s}^t \cap x$$
$$\subset N(\tilde{s}^t) \cap x$$
$$= \tilde{s}^{t+1}$$
$$\subset x$$

Now we show $\tilde{s}^{T'} \supset x$ to complete the proof. Recall the definition *partially connected*.

**Definition 1.** *We call a state $\tilde{s}'$ to be **partially connected** to state $x$, if for any cell $b \in x$, there is a cell $c_0 \in \tilde{s}' \cap x$, and a finite sequence of coordinates $c_{0:T_b}$ in $x$ that starts with $c_0$ and ends with $c_{T_b} = b$, where each subsequent element is closer than the given threshold distance $r$, i.e., for any $b \in x, \exists c_{0:T_b}$, such that $c_i \in x, d(c_i, c_{i+1}) \leq r, 0 \leq i \leq T_b$ and $c_0 \in \tilde{s}', c_{T_b} = b$.*

By definition, there exists a sequence of coordinates $c^b_{0:T_b}$ for each coordinate $b \in x$. Now we show $c^b_t \in \tilde{s}^{t'+t}$ using mathematical induction. $c^b_0 \in \tilde{s}^{t'}$. Assuming $c^b_t \in \tilde{s}^{t'+t}$ for $t > 0$,

$$c^b_{t+1} \in \mathcal{N}(c^b_t) \cap x$$
$$\subset \mathcal{N}(\tilde{s}^{t'+t}) \cap x$$
$$= \tilde{s}^{t'+t+1}$$

So $c^b_{T_b} = b \in \tilde{s}^{t'+T_b}$. If we set $T' = \max_{b \in x} T_b$, $x \subset \bigcup_{b \in x} \tilde{s}^{T_b} \subset \tilde{s}^{T'}$ holds, where the second subset holds due to the increasing property of $\tilde{s}^{t':\infty}$ proven above, i.e., $\tilde{s}^{t_1} \subset \tilde{s}^{t_2}$ for $t_1 < t_2$. Thus we conclude that $x \subset \tilde{s}^{T'}$. □

## B    FULL DESCRIPTION OF TRAINING PROCEDURE

The training procedure described in Sec. 3 introduces the infusion chain and provides high-level description of the training procedure. When the training data is sampled from the infusion chain, we utilizes the data buffer $\mathcal{B}$ as presented in experience replay (ji Lin (1992)). Sequential state transitions of the same data point, including our proposed infusion chain, are highly correlated and increase the variance of updates. With the data buffer $\mathcal{B}$, we can train the network with a batch of state transitions from different data points, and de-correlate gradients for back-propagation.

The overall training procedure is described in Algorithm 1. The buffer $\mathcal{B}$ carries tuples that consists of current state $s$, the whole shape $x$, and the time step $t$. Given the maximum budget $|\mathcal{B}|$, the buffer is initialized by the tuples $(\tilde{s}^0, x, 0)$ where $\tilde{s}^0 \sim q^0(\tilde{s}^0|x)$ for a subset of data $x \in \mathcal{D}$. For probabilistic shape completion $q^0(\tilde{s}^0|x)$ is a subset of $x$, and for shape generation we sample just one cell $\{c\} \subset x$. Then each training step utilizes $M$ tuples of mini-batch popped from the buffer $\mathcal{B}$. We update the parameters of neural network by maximizing the log-likelihood of the state that is closest to $x$ among the neighborhood of the current state, i.e., $\arg\max_\theta \log p_\theta(x \cap N(\tilde{s}^t)|\tilde{s}^t)$. Then we sample the next state from the infusion chain as defined in Eq. (3). If the next state does not meet the stopping criterion, then the tuple with the next state $(\tilde{s}^{t_i+1}_i, x_i, t_i + 1)$ is pushed back to the buffer. Else the buffer samples new data from the dataset.

While the stopping criterion (line 9) should reflect the convergence to the whole shape $x$, the amount of time steps needed to generate the complete shape varies significantly depending on the incompleteness

---

**Algorithm 1** Training GCA

1: Given dataset $\mathcal{D}$, neural network parameter $\theta$
2: Initialize buffer $\mathcal{B}$ with maximum budget $|\mathcal{B}|$ from $\mathcal{D}$
3: **repeat**
4:     Pop mini-batch $(\tilde{s}_i^{t_i}, x_i, t_i)_{i=1:M}$ from buffer $\mathcal{B}$
5:     $\mathcal{L} = 0$
6:     **for** each index $i$ in mini-batch **do**
7:         $\mathcal{L} \leftarrow \mathcal{L} + \log p_\theta(x_i \cap N(\tilde{s}_i^{t_i})|\tilde{s}_i^{t_i})$
8:         $\tilde{s}_i^{t_i+1} \sim q^{t_i}(\tilde{s}_i^{t_i+1}|\tilde{s}_i^{t_i}, x_i)$
9:         **if** $\tilde{s}_i^{t_i+1}$ does not meet stopping criterion **then**
10:             Push $(\tilde{s}_i^{t_i+1}, x_i, t_i + 1)$ into buffer $\mathcal{B}$
11:         **else**
12:             Sample $x_i \in \mathcal{D}$
13:             Sample $\tilde{s}_i^0 \sim q^0(\tilde{s}_i^0|x_i)$
14:             Push $(\tilde{s}_i^0, x_i, 0)$ into buffer $\mathcal{B}$
15:         **end if**
16:     **end for**
17:     Update parameters $\theta \leftarrow \theta + \eta\frac{\partial \mathcal{L}}{\partial \theta}$
18: **until** convergence of training or early stopping

---

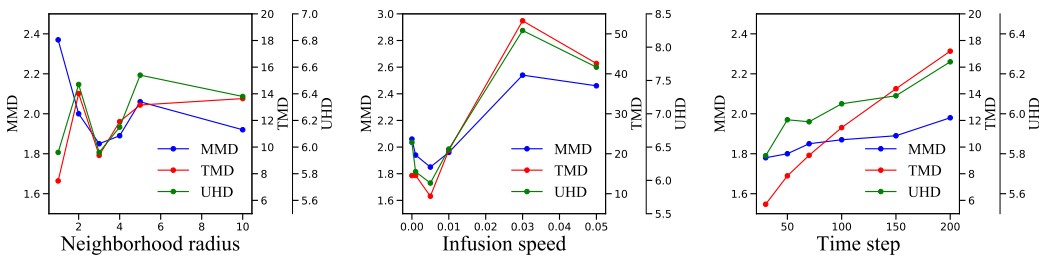

Figure 8: Ablation study on the effects of neighborhood radius $r$, infusion speed $w$, and time step $T$ (from left to right) tested with probabilistic shape completion on lamp dataset. Note that MMD (quality), TMD (diversity) and UHD (fidelity) presented in the figures are multiplied by $10^3$, $10^2$ and $10^2$, respectively.

of the initial shape, the efficiency of the stochastic transitions, and the complexity of the full shape. In addition, we need to learn to stay at the complete state once it is reached. We propose a simple solution with an adaptive number of training steps for each data: When $\tilde{s}_t$ contains 95% of the $x$ for the first time, we train only constant $\hat{T}$ more time steps. The additional $\hat{T}$ steps lead to learn a transition kernel similar to that of identity, implicitly learning to stay still after the generation is completed as in Anirudh et al. (2017).

GCA framework is successfully trained to generate a wide range of shapes even though our transition kernel is confined within its local neighborhood. Over 99% of the training data in generation and completion experiments satisfies the stopping criterion, except for lamp in PartNet-Scan dataset on shape completion, in which 97% of data satisfies the stopping criterion.

## C ABLATION STUDY ON HYPERPARAMETERS

In this section, we further investigate the effects of hyperparameters of neighborhood radius $r$, infusion speed $w$, and time step $T$, with probabilistic shape completion on lamp dataset. The default hyperparameters are set to $r = 3$, $w = 0.005$, $T = 70$, and for each ablation study, only the corresponding feature is changed. For the study on neighborhood radius and infusion speed, the networks were retrained with the matching hyperparameters, while the ablation study on time steps show the result of the model trained on default hyperparameter but tested with the variation of time

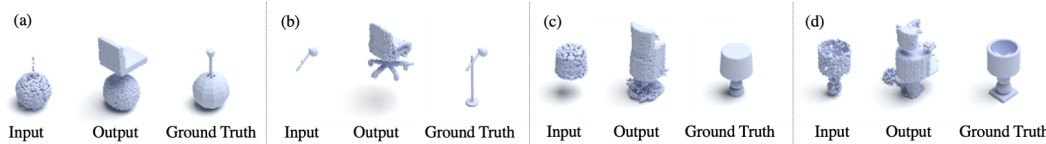

Figure 9: Completion results on unseen initial states. The model is trained on chair, but the initial state of lamp is given as input.

step $T$. The Figure 8 shows the effects of $r$, $w$ and $T$ on MMD (quality), TMD (diversity), and UHD (fidelity).

**The effect of the neighborhood radius $r$.** The neighborhood radius $r$ is the size of update neighborhood of the probabilistic distribution for individual cells, $\mathcal{N}(c_i) = \{c' \in \mathbb{Z}^n |\ d(c_i, c') \leq r\}$. When the radius $r$ is too small ($r = 1, 2$), the MMD (quality) is considerably high compared to when $r = 3, 4$. Since the size of update is small, models require more time steps to reconstruct the whole shape, and are not able to generate the disconnected lamps as shown in Figure 7. When the neighborhood size is adequate ($r = 3, 4$), GCA produces fine reconstructed shapes with reasonable scores on all metrics. However, when the radius is too high ($r = 5, 10$), the model yields shapes that are considerably irrelevant to the given initial state, resulting in a high TMD and UHD scores. Due to the large possible transition states, few bad samplings lead to states that are irrelevant to the initial state, resulting in the degradation of performance.

**The effect of the infusion speed $w$.** The infusion speed controls how likely the groudtruth shape is injected ($\alpha^t = \max(wt, 1)$ in Eq. (3)) with the increasing time step. When the infusion speed is too low, the infusion chain is close to the sampling chain, but it is less likely to visit a state that observes the whole complete state. Also, as Huszár (2015) said, the model learns a biased estimator, which implies that GCA may not converge to the correct model. The results in Figure 8 show that MMD and UHD are high when the infusion speed $w = 0, 0.001$ compared to when $w = 0.005$. This indicates that small $w$ generates shapes with worse quality and fidelity. On the other hand, when the infusion speed is set too high ($w = 0.03, 0.05$), the discrepancy between the infusion chain and sampling chain increases. We noticed the diverging behavior of GCA, when the infusion speed was set too high. Bordes et al. (2017) state that the optimal value for the infusion speed differs depending on the time step $T$. We empirically saw that $w = 0.005$ produced the best result for both shape generation and completion with $T = 100$ and $T = 70$.

**The effect of the number of time step $T$.** The time step $T$ indicates the number of state transitions performed to generate the shape $s^0 \rightarrow s^1 \rightarrow ... \rightarrow s^T$. The right plot of Figure 8 shows the performance of GCA with different time steps trained on the default hyperparameter. Even though MMD is lowest when T=30, some lamps fail to complete the shape in the given time steps. As the time steps increase, all scores tend to increase by morphing the initial state to the complete state. However, we would like to emphasize that the scale of the plot is relatively small and the model is able to keep the MMD (quality) below 0.002 even when ran on 200 time steps. This is substantially low considering that the MMD of cGAN (Wu et al. (2020)) is 0.00197. This demonstrates that GCA is able to maintain the quality of shapes, being robust to increasing time steps.

## D   COMPLETION RESULTS ON UNSEEN INITIAL STATES

We investigate the behavior of GCA when an unseen input is given as the initial state. The Figure 9 shows the result of GCA trained on chair completion, but a partial lamp is given as the input. The behavior can be classified largely into three classes. First, the model respects the initial state and is able to generalize to a novel shape as in Figure 9 (a). Second, as in Figure 9 (b) and (c) which are the most common cases, the model deforms the initial state into a trained state and generates results learned during training. The final state contains a state similar to the initial, but deformed to a shape that belongs to the trained category. Lastly, like in Figure 9 (d), if the initial state is dissimilar to the visited state, it fails to generalize and starts diverging.

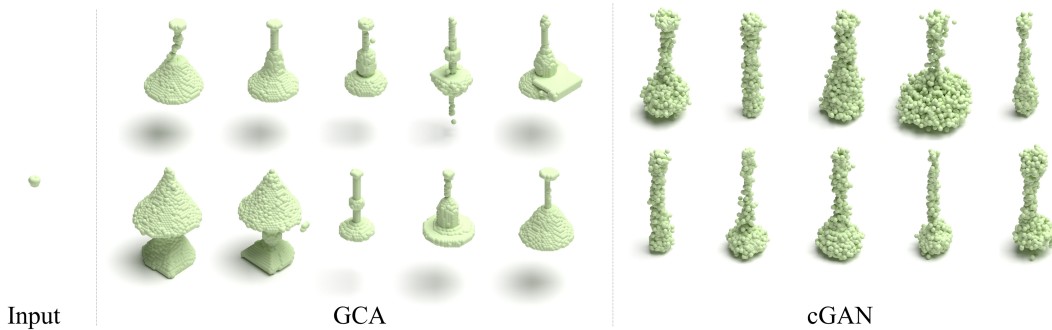

Input           GCA           cGAN

Figure 10: Qualitative comparison against cGAN (Wu et al. (2020)) for lamp with small partial input.

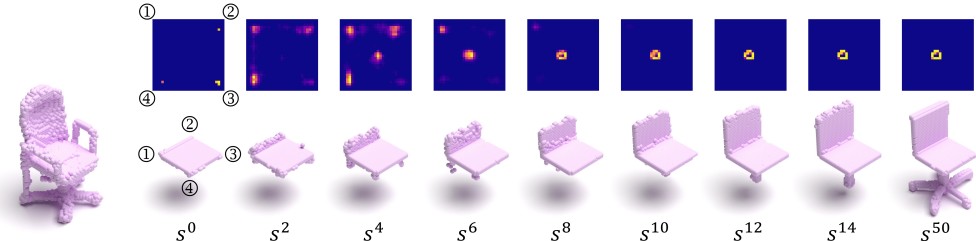

Figure 11: Figure 12: Probability heatmap of chair completion. The probability of next state
Failure case. of occupancy is presented at a horizontal cross section, immediately below the
input chair seat. We also present the current state $s^t$.

## E    DIVERSITY ANALYSIS ON LAMP DATASET

Among the results presented in Sec. 4, our diversity score (TMD) for shape completion is significantly higher than the state-of-the-art with lamp dataset (Table 1). By further investigating instances with high TMD scores in lamp dataset, we could observe that the input part is small and highly incomplete. The input partial data is created by randomly selecting a part from the instance, and the part labels in lamp dataset are fragmented and diverse with less regular structure with many small parts. A typical example with high TMD score is shown in Figure 10. The input is a small part of a lamp, and our completion results cover a wide variety of lamps including table lights and ceiling lights, and still exhibit fine geometry. In contrast, cGAN (Wu et al. (2020)) handles the ambiguity with consistently blurry output. The TMD score for this specific example is 20.45 for GCA, while the cGAN achieves 6.592.

## F    CONVERGING TO A SINGLE MODE

Our scalable formulation can generate multiple voxels at a single inference time. This is a unique characteristic of our formulation compared to other autoregressive models (van den Oord et al. (2016); Sun et al. (2020)) and the key to handle high-resolution 3D shapes. Specifically, the multiple voxels of the next state are sampled from a conditionally independent occupancy for each cell, i.e., $p_\theta(s^{t+1}|s^t) = \prod_{c \in \mathcal{N}(s^t)} p_\theta(c|s^t)$. While we gain efficiency by making multiple independent decisions in a single inference, there is no central algorithm that explicitly controls the global shape. In other words, we can view the generation process of GCA as morphogenesis of an organism, but there is no unified 'DNA' to eventually move towards a single realistic shape. We indeed encounter rare failure cases that make multiple conflicting decisions on the global structure, as presented in Figure 11.

However, for most of the cases, our model is capable of generating a globally consistent shapes, converging to a single mode out of multi-modal data distribution. For example, when a chair seat is given as the partial input shape as in Figure 12, there are multiple possible modes of shape completion, including four-leg chairs or swivel chairs. The ambiguity is present in $s^{2:6}$, where both the center and

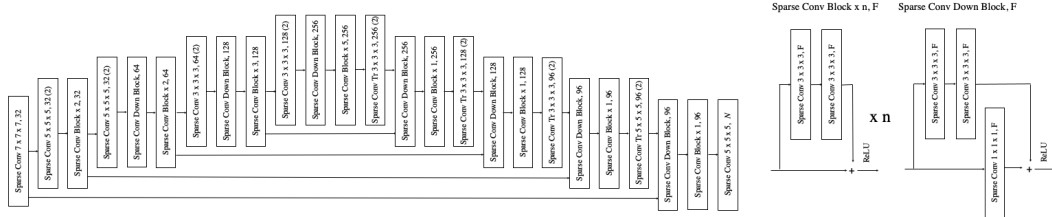

Figure 13: Architecture of U-Net. The parenthesis denotes the stride of the sparse convolution / deconvolution. Every convolution is followed by batch normalization.

the corners have high probability of chair legs. GCA eventually picks a mode and decides to erase the legs at the corners and only grow towards the center, creating a swivel chair. As shown in this example, the transition kernel of GCA observes the change in the context and eventually converges to a globally consistent mode even with independent samples of individual cells.

## G EXPERIMENTAL DETAILS

### G.1 NEURAL NETWORK ARCHITECTURE AND IMPLEMENTATION DETAILS

All experiments conducted use a variant of U-Net (Ronneberger et al. (2015)) architecture implemented with sparse convolution library MinkowskiEngine (Choy et al. (2019)), depicted in Figure 13. We train our method using the Adam (Kingma & Ba (2015)) optimizer with an initial learning rate of 5e-4 and batch size of 32. The learning rate decays by 0.5 every 100k steps. All experiments are run on RTX 2080 ti 11GB GPU, except for the analysis on the neighborhood size with $r = 10$, which ran on Titan RTX 24GB GPU.

### G.2 BASELINES

All experiments in Table 1 and Table 2 are excerpted from Wu et al. (2020) and Cai et al. (2020) except for the results of 3D-IWGAN (Smith & Meger (2017)). 3D-IWGAN was trained on the dataset same as ours for 4000 epochs using the author's offical code: `https://github.com/EdwardSmith1884/3D-IWGAN`, with the default hyperparameters. Since the original code only supports 32 x 32 x 32 voxel resolution, we added extra convolution / deconvolution layer for the GAN's discriminator, generator and VAE, generating 64 x 64 x 64 voxel resolution.

### G.3 EVALUATION METRICS

We provide a more detailed explanation regarding quantitative metrics for evaluation of our approach. Since we compare our method against recent point cloud based methods, we convert voxels into point clouds simply by treating the coordinates of occupied voxels as points.

For all of the evaluations, we use the Chamfer distance (CD) to measure the similarity between shapes represented as point clouds. Chamfer distance between two sets of point cloud $X$ and $Y$ is formally defined as

$$d_{CD}(X, Y) = \sum_{x \in X} \min_{y \in Y} \|x - y\|_2^2 + \sum_{y \in Y} \min_{x \in X} \|x - y\|_2^2. \tag{4}$$

**Probabilistic Shape Completion**. We employ the use of minimal matching distance (MMD), total mutual difference (TMD), and unidirectional Hausdorff distance (UHD), as in Wu et al. (2020).

Let $S_p$ be the set of input partial shapes and $S_c$ be the set of complete shapes in the dataset. For each partial shape $P \in S_p$, We generate $k = 10$ complete shapes $C_{1:k}^P$. Let $G = \{C_{1:k}^P\}$ be the collection of the entire generated set started from all elements in $S_p$.

- **MMD** measures the quality of the generated set. For each complete shape in the test set, we compute the distance to the nearest neighbor in the generated set $G$ and average it, which is

formally defined by

$$\text{MMD} = \frac{1}{|S_c|} \sum_{Y \in S_c} \min_{X \in G} d_{CD}(X, Y). \tag{5}$$

- **TMD** measures the diversity of the generated set. We average the Chamfer distance of all pairs of the generated set for the same partial input, formally defined by

$$\text{TMD} = \frac{1}{|S_p|} \sum_{P \in S_p} \frac{2}{k(k-1)} \sum_{1 \leq i < k} \sum_{i < j \leq k} d_{CD}(C_i^P, C_j^P). \tag{6}$$

- **UHD** measures the fidelity of the completed results against the partial inputs. We average the unidirectional Hausdorff distance $d_{HD}$ from the partial input to each of the $k$ completed results, formally defined as

$$\text{UHD} = \frac{1}{|S_p|} \sum_{P \in S_p} \frac{1}{k} \sum_{1 \leq i \leq k} d_{HD}(P, C_i^P). \tag{7}$$

**Shape Generation**. We employ the use of 1-nearest-neighbor-accuracy (1-NNA), coverage (COV, and minimal matching distance (MMD)) as in Cai et al. (2020). Since MMD was introduced above, we state the definition of 1-NNA and COV. Let $S_g$ be the set of generated shapes and $S_r$ be the set of reference shapes.

- **1-NNA**, proposed by Lopez-Paz & Oquab (2016), evaluates whether two distributions are identical. For shape $X$, we denote the nearest neighbor as $N_X = \arg\min_{Y \in S_{-X}} d_{CD}(X, Y)$, where $S_{-X}$ represents the set including the entire generated and reference shapes except itself, $S_{-X} = S_r \cup S_g - X$. 1-NNA is defined to be

$$\text{1-NNA}(S_g, S_r) = \frac{\sum_{X \in S_g} \mathbb{1}_{N_X \in S_g} + \sum_{Y \in S_r} \mathbb{1}_{N_Y \in S_r}}{|S_g| + |S_r|}, \tag{8}$$

where $\mathbb{1}$ is an indicator function. The optimal value of 1-NNA is 50%, when the distribution of two sets are equal, unable to distinguish the two sets.

- **COV** measures the proportion of shapes in the reference set that are matched to at least one shape in the generated set, formally defined by

$$\text{COV}(S_g, S_r) = \frac{|\{\arg\min_{Y \in S_r} d_{CD}(X, Y) | X \in S_g\}|}{|S_r|}. \tag{9}$$

## H   ADDITIONAL SAMPLES ON PROBABILISTIC COMPLETION

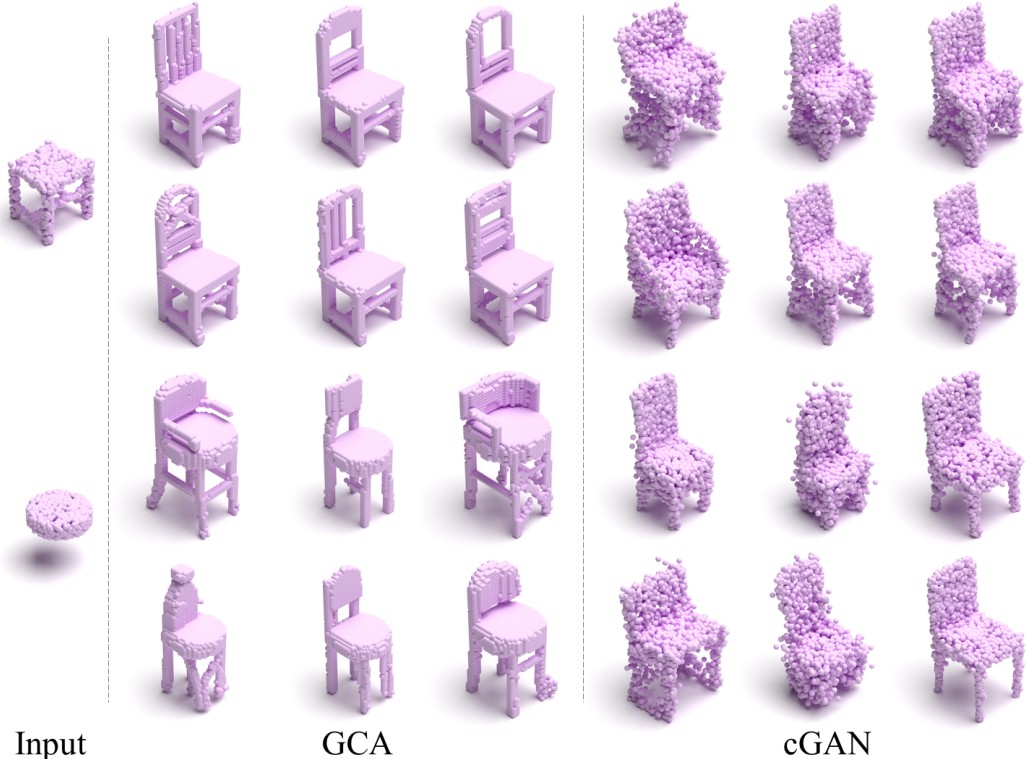

Figure 14: Qualitative comparison against cGAN (Wu et al. (2020)) of probabilistic shape completion on chair.

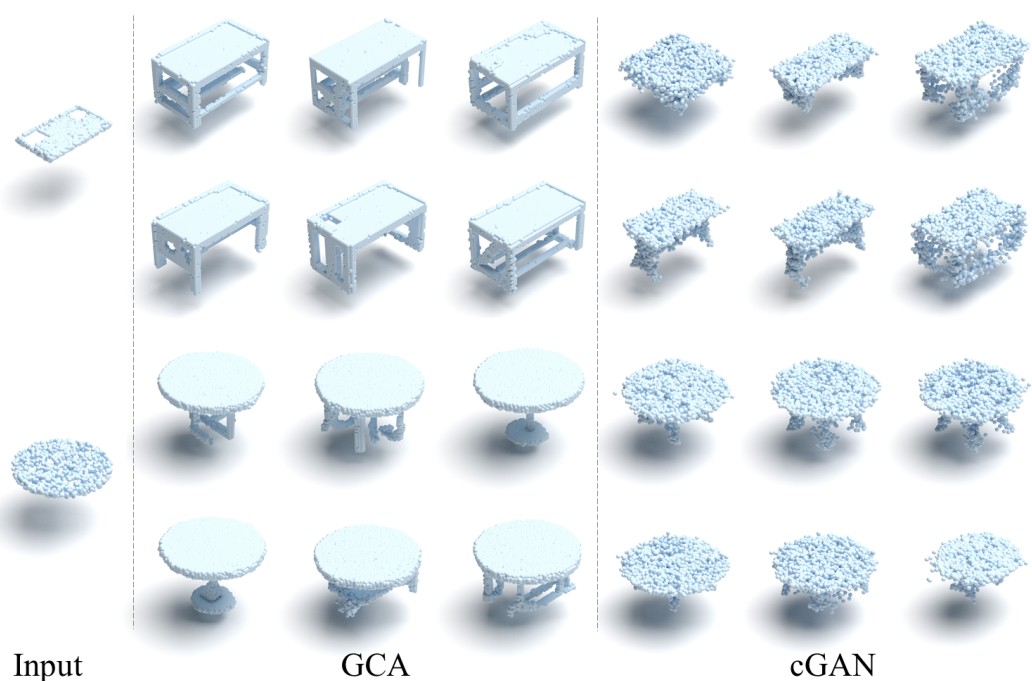

Input                          GCA                          cGAN

Figure 15: Qualitative comparison against cGAN (Wu et al. (2020)) of probabilistic shape completion on table.

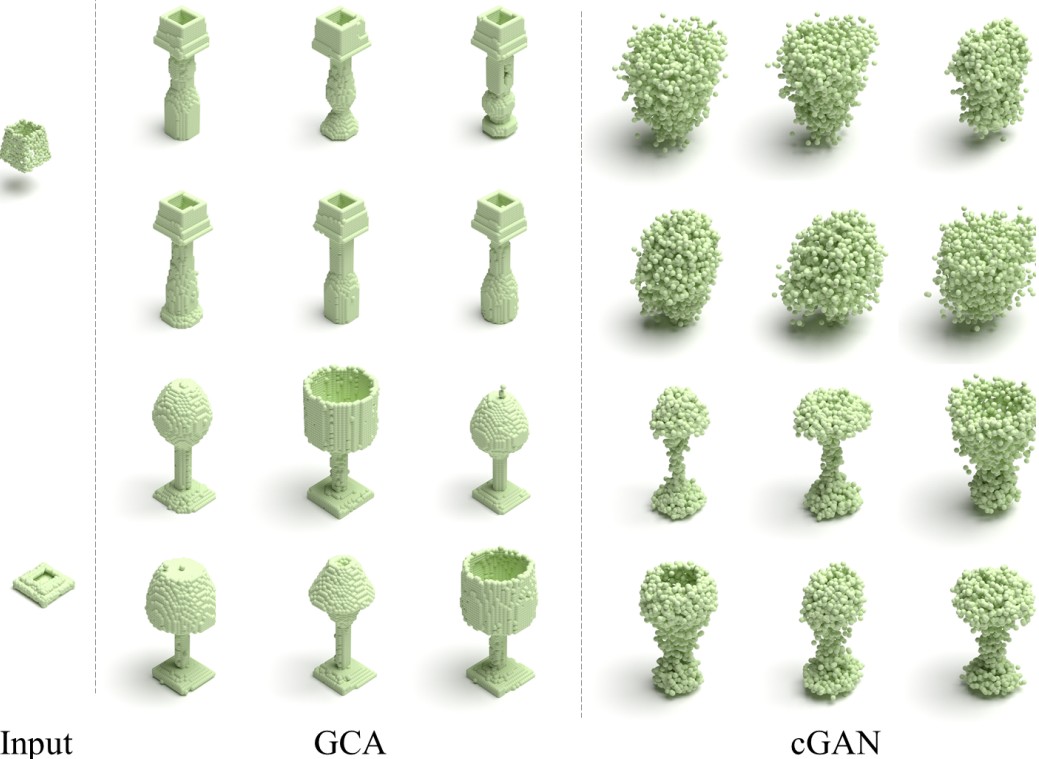

Input                          GCA                          cGAN

Figure 16: Qualitative comparison against cGAN (Wu et al. (2020)) of probabilistic shape completion on lamp.

# I ADDITIONAL SAMPLES ON SHAPE GENERATION

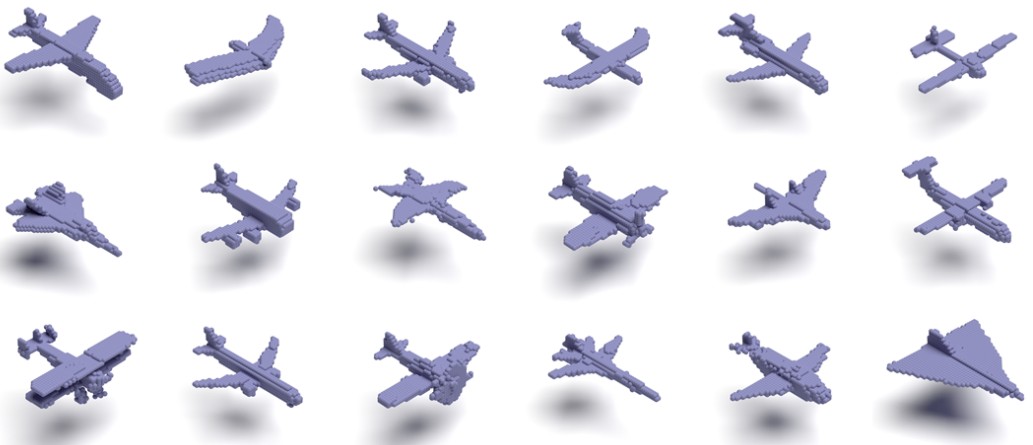

Figure 17: Samples from shape generation on airplane dataset.

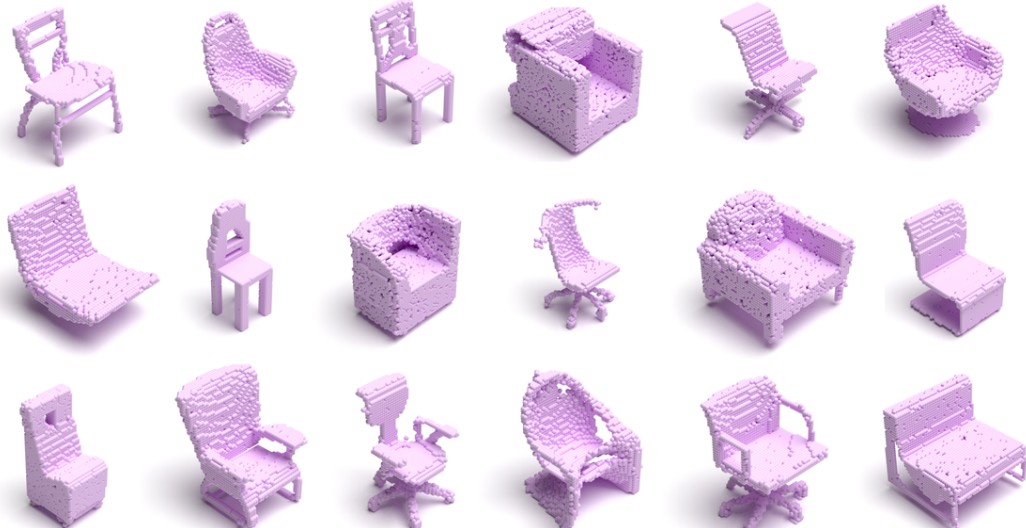

Figure 18: Samples from shape generation on chair dataset.

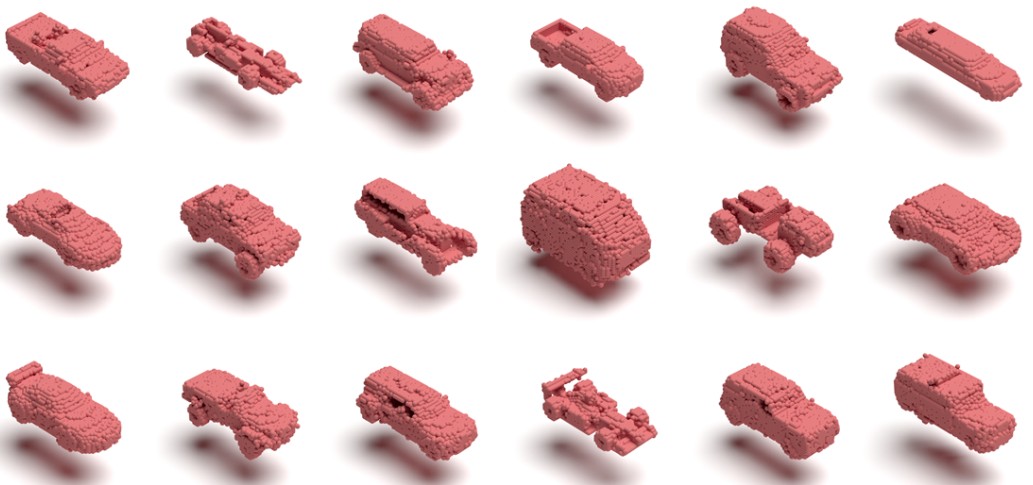

Figure 19: Samples from shape generation on car dataset.

