# OpenReview forum: "Learning to Generate 3D Shapes with Generative Cellular Automata"
_ICLR.cc/2021/Conference — ICLR 2021 Poster_

### Official Review · AnonReviewer3 · 2020-10-21
**A nice idea with minor flaws**

**Rating:** 7
**Confidence:** 3

**Review:**

The paper proposes a generative method for 3D objects (voxels representation). Given an initial voxels configuration (e.g. partial shape, or even a single voxel), the method learns a local transition kernel for a Markov chain to decide how to evolve the configuration; sampling iteratively from these probabilities leads to a final model. The paper shows results on shape completion and generation, obtaining fairly good results.

PROS
- Novelty: I find the approach interesting and the idea seems new in the 3D domain to me (while, as cited in the paper, there are some other similar works in the 2D). It may serve as inspiration for future works to build on it.

- Theoretical foundation: I like the theory on which the paper is based on. I think several properties can be obtained thanks to the theory of finite-state automata (like the presented theorem about the final convergence of the method).

CONS
- Input/Output: the method is restricted to work in the voxel setting and it is not easily extendible on point clouds. This limits the quality of the generated objects. Also, looking at the attached .gif files, it seems that in the end the model has some noise and it is hard to decide when to stop it (while it seems structural stable). Finally, while in some case the process seems to respect the initial suggestion (e.g. for the lamp_completion example), sometimes seems it does not (e.g. the car_generation start from a place and move to another; is this linked by a strange position of the initial voxel? Also the disconnected_lamp example modifies its input state)

- Results/Analysis: The experiments are a bit limited since they only involve a few rigid categories. I think the paper could be strengthened by testing the method on some non-rigid domains, e.g. humans. Also, the method requires the choice of several hyperparameters, and this is briefly discussed in Section 4.3. I would suggest performing ablation on them.
Finally, an interesting analysis could investigate how the network behaves when an "unlikely" starting state has been feed (e.g. sparse noise, or never seen parts). Is it able to recover a reliable shape?

- Presentation: some concepts (like the Cellular Automata or infusion chains) are only marginally discussed; I would like to see a better introduction to them since they are not so common concepts for Computer Graphics community. Also, the bibliography needs a serious revision, since many voices either break the margins with URLs or are incomplete (e.g. Huszar 2015).

PRE-REBUTTAL RATING

I think the paper could be strengthened by further analysis, experiments, and presentations fixing. However, I like the general idea and I am favorable to accept it as the first step of a different perspective for 3D generative methods. For the final version, I suggest including some further analysis on hyperparameters and initial-state.

-------------

FINAL RATING

I have carefully read the other reviews and authors' replies.
I agree with other reviewers that the rebuttal is satisfactory, and raised the score accordingly.
Thanks to the authors for their effort.

---

> ### Author Response · Authors · 2020-11-22
> **Reply to reviewer3**
>
> We would first like to thank the reviewer for your helpful comment. We have revised the bibliography and added further explanation of infusion chain and cellular automata in the related works section. We think our work can be extended to point cloud by combining the techniques discussed in GRNet [1], but we leave it as future work. During the rebuttal period, we have added some results on hyperparameters and the GCA’s behavior on the initial state.
>
> &nbsp;
>
> 1. Analysis of hyperparameters
>
> We have conducted experiments regarding hyperparameters (neighborhood radius r, infusion speed w, and time step T) in the appendix C.
>
> We summarize the results for the readers convenience.
> - Radius r is best to be as small as possible to reduce the chance of visiting unlikely states and forgetting the initial state. However, sufficiently big (r=3, 4) to deal with disconnected shapes.
> - Infusion speed w depends on time step T, but setting the rate to 0.005 was good for all experiments which used T = 100 and T = 70.
> - Time step T is quite robust. It was set empirically to be sufficient for GCA to finish completion and the result does not deviate significantly with different T. The longer the time steps were set, the more diverse the results, but the further it deviated from the initial state. Note that running much longer steps than the model was trained still resulted in a lower UHD (fidelity) compared to cGAN [2].
>
> &nbsp;
>
>
> 2. Behavior on unseen initial states
>
> We have added results in appendix D regarding the behavior of unseen initial states by starting in a initial state of lamp, trained on chair completion dataset. The behavior is classified in 3 ways, where the 1) model generalizes and creates a novel chair while being loyal to the input 2) the model slightly mends the input and creates a chair that it observed, 3) creating noise since and fails to generalize. 2) were observed as the most common behavior.
>
> &nbsp;
>
> 3. Regarding gif files
>
> We explain the phenomenon of sometimes erasing the initial suggestion.
>
> As added in the related works section, the heart of infusion training lies on the diffusion based models, which is basically denoising model. Since GCA was trained with infusion training, it can be viewed as a part of denoising model. When learning with infusion training, GCA does not know what part of input state is the initial suggestion (due to the Markov property of states). Thus every voxel that comes to input has always the ambiguity of being erased. In the PartNet dataset, the initial suggestion is always considered to be accurate (part of the complete data), thus erasing the initial suggestion would cause a negative effect. However, when considering the use of GCA in the real scans, the input is not always guaranteed to be part of the ground truth and can be noise. Thus, we think that denoising of the input can be considered positive, if it happens up to some extent. We believe that the UHD (fidelity) score of the completion experiment shows the GCA is able to denoise the initial state up to a reasonable amount.
>
> If we are confident that the initial state is a perfect state, it would be an interesting approach to mark the initial voxels as a new state (other than just occupied or not) and perform the same training.
>
>
> &nbsp;
>
>
> 4. Dataset
>
> The results reported in the paper follow the conventional experimental setup for generative models (pointflow [3], shapegf [4]) and probabilistic completion models (cGAN [1]).
>
> In addition, we added additional experiments suggested by R1, and tried a) training for entire category, b) training from one class and completing for another class, c) compleing the scene with multiple objects. The results are included in Figure 5 and appendix D.
>
> The extension into more general setup including non-rigid shape is very interesting. Especially we would like to extend the work into dynamic deformation or more general set up with control parameters in the future.
> &nbsp;
>
> [1] Xie et al., GRNet: Gridding Residual Network for Dense Point Cloud Completion, ECCV (2020)
> [2] Wu et al., Multimodal Shape Completion via Conditional Generative Adversarial Networks, ECCV (2020)
> [3] Yang et al., Pointflow: 3d point cloud generation with continuous normalizing flow, ICCV (2019)
> [4] Cai et al., Learning gradient fields for shape generation, ECCV (2020)

---

### Official Review · AnonReviewer4 · 2020-10-26
**Strong paper but needs voxel comparison**

**Rating:** 8
**Confidence:** 5

**Review:**

I think this is a strong and interesting paper. It is well written and properly explains its ideas. I am especially impressed by the figures which are clean and helpful in understanding the ideas of the paper and performance of the method. I am also am happy with the variety of experiments, which answered the questions which arose to me when understanding the method.

My only real concern with respect to this paper is the lack of comparison to highly relevant papers . It does not compare to any other voxel-based 3D generation methods, only to point cloud ones. I think this is quite an important sticking point which I strongly encourage the authors to address, as without comparison within the same 3D representation it is very difficult to establish how much of the claimed improvement in performance is directly due to the benefits of the chosen representation. A clear example of this is in the visual comparison to cGAN (Figure 3) where your generations look far cleaner, however, this might just be a consequence of predicting in a grid as opposed to in a continuous space.  For 3D generation it would be a good idea to compare to : "Learning a Probabilistic Latent Space of Object Shapes via 3D Generative-Adversarial Modeling" ,  and "Improved Adversarial Systems for 3D Object Generation and Reconstruction". For shape completion you write: "we report cGAN as the baseline to our method, since this is the only approach announced in probabilistic shape completion". This is just not true. For example, a you could compare again to "Improved Adversarial Systems for 3D Object Generation and Reconstruction", and also to "3D Object Reconstruction from a Single Depth View with Adversarial Learning". I would like to see experiments comparing to other voxel based methods in the paper, and also shown here.  I would be very happy to raise my score if this is provided.  At the very least other voxel based methods should be referenced, and their absence from the experimental comparison justified.

---

> ### Author Response · Authors · 2020-11-22
> **Reply to reviewer4**
>
> We appreciate the reviewer for your thoughtful comment.
>
> The main reason we compared our method with point cloud representations was due to the recent focus on 3D shape generation used point cloud based representation rather than the voxel counterpart. We realize that comparing voxel based generation is a great way to show that the performance gain regardless of the representation and conducted experiments comparing our method against voxel based methods to relieve your concern.
>
> During the rebuttal period, we’ve tried comparing our method against “Improved Adversarial Systems for 3D Object Generation and Reconstruction” (3D-IW-GAN) for generation and probabilistic completion. We also tried to compare our method against “Learning a Probabilistic Latent Space of Object Shapes via 3D Generative-Adversarial Modeling” using a repo with most stars (https://github.com/rimchang/3DGAN-Pytorch), but the training failed due to the severe instability of GAN training in 64 x  64 x 64 grid space. All the results are reported in the Table 1 and Table 2 of our manuscript.
>
> For 3D-IW-GAN, we used the official repository (https://github.com/EdwardSmith1884/3D-IWGAN) with default hyperparameters and trained for 4000 epochs for completion and generation with the same dataset as ours. Since the repo only supported 32 x 32 x 32, we added another convolution / deconvolution layer for the generator, discriminator and the VAE to compare on 64 x 64 x 64 voxel space.  We would also like to mention that the scores of our method changed due to hyperparameter selection as R1 mentioned, and longer training for generation, which improved our MMD score. The results are highlighted for your convenience as follows.
>
> - Probabilistic completion:
>
> |          | MMD   |      |       | TMD   |      |       | UHD   |      |       |
> |----------|-------|------|-------|-------|------|-------|-------|------|-------|
> |          | Chair | Lamp | Table | Chair | Lamp | Table | Chair | Lamp | Table |
> | 3D-IWGAN | 1.94  | 3.57 | 8.18  | 0.99  | 3.58 | 1.52  | 6.89  | 8.86 | 8.07  |
> | GCA      | 1.28  | 1.85 | 1.13  | 4.74  | 9.38 | 4.50  | 6.21  | 5.96 | 5.60  |
>
> Note that for MMD and UHD, lower is better and for TMD, higher is better. Our method outperforms the 3D-IWGAN in all of the scores, which implies that GCA can generate high fidelity (MMD) with diverse shapes (TMD) while being loyal (UHD) to the given partial geometry. We observed that 3D-IWGAN generated consistent shapes for the input, which resulted in a low TMD and were especially weak at generating thin objects (lamps, and leg of tables).
>
> During the revision period, we have added a few experiments in probabilistic completion where all classes (chair, lamp, table) were jointly trained as asked by R1. Still our method outperforms all other methods on MMD and TMD in probabilistic completion in terms of all classes on average. We have also conducted experiments where we can complete a scene consisting of multiple partial objects, where the GCA was only trained on a single object.
>
>
> - Shape generation:
>
> |          | 1-NNA    |       |       | COV      |       |       | MMD      |      |       |
> |----------|----------|-------|-------|----------|-------|-------|----------|------|-------|
> |          | airplane | car   | chair | airplane | car   | chair | airplane | car  | chair |
> | 3D-IWGAN | 94.70    | 76.60 | 63.62 | 38.02    | 42.90 | 45.17 | 1.53     | 4.33 | 15.71 |
> | GCA      | 90.62    | 65.06 | 65.71 | 35.80    | 46.31 | 44.56 | 1.25     | 4.19 | 16.89 |
>
> Note that for 1-NNA and MMD, lower is better and for COV, higher is better. 3D-IWGAN were surprisingly good at generating chair category and were comparable to SOTA methods. However, for airplane and car category, the method tends to generate blurry objects that resulted in a bad MMD and 1-NNA score compared to GCA.

---

> > ### Comment · AnonReviewer4 · 2020-11-23
> > **Strong paper**
> >
> > I believe this has properly addressed my concern, and the paper is stronger for it. I have raised my scored accordingly.

---

### Official Review · AnonReviewer1 · 2020-10-27
**A good submission with promising results, but needs some more clarifications**

**Rating:** 6
**Confidence:** 5

**Review:**

This paper proposes an interesting formulation of 3D object point cloud generation -- sampling from the transition kernel of a Markov chain. Given an input partial shape, the method iteratively updates the 3D shape by growing point clouds in a local region, and finally outputs the complete 3D object. The proposed method is interesting and technically sound. The quantitative result achieves the state-of-the-art for shape completion and shows comparable results for shape generation. Besides, the paper is well written and easy to follow.
My detailed comments are as follows.
1. It would be better to evaluate the inference speed compared with the state-of-the-arts. I'm curious about the computation cost of the Markov chain based 3D generative model.
2. Although I like the illustration shown in Figure 6, I am still worried about whether different hyper-parameter sets (r, T, and infusion speed) will affect performance. For the example in Figure 6, it would be better to use different hyper-parameters to show qualitative comparisons. I'm also a bit concerned about the generalization ability of this approach, since the proposed method uses different sets of hyper-parameters for chair, table and lamp as stated in Sec.4.1, the first paragraph. It would be better to show the quantitative results using the same set of hyper-parameters for all classes.
3. For the experiment (shape completion and shape generation), is the model trained for all categories or for each category respectively? Will different training settings affect the overall performance? My intuition is that this method can complete 3D objects (for example, only given the bottom of the lamp), because it is trained from a large amount of data, so it is easy to "remember" the data distribution of each class. If the model is trained on a single category, it is not surprising that it performs well. I want to know whether the model itself can learn categorical features through joint training of all classes. It will also be interesting to see that the shape generation result is a fusion of several classes -- this can show encouraging results for applications such as 3D object design.
4. It would be nice to discuss how to extend this method to use other representations (for example, voxels, meshes, or implicit functions) for 3D object generation.

In general, I like the proposed idea and the results are promising. However, considering my above concerns (mainly #2 and #3), I think this article is "slightly above the threshold". I will re-evaluate the score based on the author’s feedback, .

---

> ### Author Response · Authors · 2020-11-22
> **Reply to reviewer1**
>
> Thank you for the constructive and encouraging comments.
>
>
> 1.
>
> GCA takes 0.7 seconds on average wall-clock time to generate one lamp completion (time of 70 inferences combined). Admittedly GCA is slower than recent methods since it is a Markov chain based method. However, we think another bottleneck is the slow hashing operation of sparse convolution library (MinkowskiEngine [1]), which is considerably slow compared to normal convolutions. To increase the speed of the sparse convolution, major progress has been made recently (such as using gpu-hashtable). By employing the efficient version, we think there will be an additional speed boost that GCA will directly inherit.
> &nbsp;
>
> 2.
>
> We have updated the results for Table1 (completion results) for all classes using the same hyperparameter r=3, T=70, infusion speed=0.005, the hyperparameter we used for the lamp class. There was a significant performance improvement in diversity (TMD) for chair and table class, with a slight degradation in MMD (quality) and UHD (fidelity to input). We would like to note that there was no extensive search in the hyperparameter before submission on chair and table category.
>
> The changes on the chair and table partnet dataset are as follows:
>
> |                  | MMD  | TMD  | UHD  |
> |------------------|------|------|------|
> | Chair (previous) | 1.26 | 1.79 | 6.10 |
> | Chair (updated)  | 1.28 | 4.74 | 6.21 |
> | Table (previous) | 1.06 | 2.04 | 5.45 |
> | Table (updated)  | 1.13 | 4.50 | 5.60 |
>
>
> We have also conducted an additional ablation study in appendix C, where we investigate the results with different hyperparameters for the probabilistic shape completion of the lamp dataset.
> &nbsp;
>
> 3.
>
> The results reported in the paper are all separately trained in a single class. This has been the conventional experimental setup for generative models (pointflow [2], shapegf [3]) and probabilistic completion models (cGAN [4]).
>
> However, we were also intrigued with the review, and conducted an experiment on whether our model is able to learn from joint training in all classes and benefit from it. It turns out we can train the GCA without the major expense of performance! We’ve attached an additional Figure 5 in the manuscript to show that our model is able to generate shapes that have mixed features of each other. We also added the performance of joint trained model in Table 1, and there was no significant performance degradation on chair and table dataset compared to the individually trained model.
>
> From your thoughtful review, we were also curious whether the GCA could be extended to complete a scene consisting of multiple objects, since the model was already translation invariant. GCA was able to complete a whole scene if the objects were distant enough as in Figure 5, even though it was trained with a single object. We would like to note that no special training was made and the effect on distant voxels was rather small.
> &nbsp;
>
> 4.
>
> For point cloud representation, we think our method can be extended by using the same techniques as in GRNet [5], which uses voxel as intermediate representation for point cloud completion. We are also positive towards combining our method with PolyGen [6], which creates mesh shape in an autoregressive fashion. Instead of generating one vertex at a time, we think mixture of our “parallel” generation from GCA can help speed up the method.
>
> [1] Choy et al., 4d spatio-temporal convnets: Minkowski convolutional neural networks, CVPR (2019)
> [2] Yang et al., Pointflow: 3d point cloud generation with continuous normalizing flow, ICCV (2019)
> [3] Cai et al., Learning gradient fields for shape generation, ECCV (2020)
> [4] Wu et al., Multimodal Shape Completion via Conditional Generative Adversarial Networks, ECCV (2020)
> [5] Xie et al., GRNet: Gridding Residual Network for Dense Point Cloud Completion, ECCV (2020)
> [6] Nash et al., PolyGen: An Autoregressive Generative Model of 3D Meshes, ICML (2020)

---

> > ### Comment · AnonReviewer1 · 2020-11-23
> > **Marginally above acceptance threshold**
> >
> > I appreciated the response from the authors. After reading the rebuttal and feedback from other reviewers, I vote for accept. The authors have addressed most of my concerns.
> >
> > But I would stick to my original rating as "marginally above acceptance threshold". I want to point out that during the rebuttal period, other experiments/adjustments to further improve the proposed method may not be considered. This is to ensure fairness to other paper authors. Therefore, even if the proposed method becomes better and stronger after parameter tuning and extended experiments, I will not increase the score.

---

### Author Response · Authors · 2020-11-12
**Authors' response**

We thank all the reviewers for the insightful and considerate suggestions that will improve our paper. We will soon update the paper and answer to each and every review with a dedicated response.

---

### Public Comment · ~Alexander_Mordvintsev1 · 2020-11-12
**Relation to Growing Neural CA work**

The objective of finding a uniform iterative update rule, that builds a complete pattern from a single seed cell or a damaged pattern, makes this paper look similar to the [Growing Neural CA](https://distill.pub/2020/growing-ca/), published in February 2020 at Distill.pub. The dynamics of the shape generation process looks very similar as well. Below I’ll try to highlight the difference between this work (3DGCA) and “Growing Neural CA” (NCA).

3DGCA tries to train a probabilistic incremental update process that would allow (conditional) sampling from a distribution of 3D models. Randomness of the updates is used as a source of variation of the generated 3D shapes. NCA paper focuses on training the incremental update cell rule that would build a pixel-perfect copy of a predetermined pattern, starting from a seed cell, or a damaged state. The stochasticity of cell updates is considered to be an adverse factor that the model needs to be robust against.

NCA determines starting and desired final conditions of the grid, and uses back-propagation through time to find the iterative update rule. 3DGCA uses infusion chain sampling to construct synthetic series of grid states.

NCA model is a classical CA, where each cell only has local information about its 3x3 neighbourhood, and performs updates to its own self. Cells have to rely on 16-dimensional state vectors to coordinate the process of pattern formation. The premise is modelling the natural embryogenesis process of global self-organization through local communication.

3DGCA cells can only be in one of the two states: empty or occupied, but the update rule is very powerful, and, in my opinion can be called a Cellular Automaton with quite a bit of a stretch. In particular, the “neighbor size” parameter only determines the “voting neighborhood” of a cell, but the “perception neighborhood” is not specified in the paper. We only know that the update rule uses a “variant of U-Net”, which implies multiscale information propagation through pooling and convolution operations. This way the information, available for each cell, can be gathered over a very large volume, potentially covering the whole growing pattern. Technically this can still be called a CA due to the uniformity of the update rule, but to me it feels a bit of a stretch when every cell can be considered a neighbour (i.e. influence the decision) of every other cell.

I’d recommend mentioning Growing NCA work in the related work section, and clarifying the details of the proposed model with respect to the classical CA definition.

---

> ### Comment · AnonReviewer3 · 2020-11-13
> **Reply**
>
> Thanks, Alexander; indeed NCA seems relevant, and your comparison between the two works is really useful.
> It seems there are crucial differences that distinct the two works (the relation with the randomness and the aim of the generation are the most relevant to me). I agree with you that NCA should be referenced, and a comment on the main differences in the technical choices would be beneficial.

---

> ### Author Response · Authors · 2020-11-14
> **Reply**
>
> Thank you for the comments, and we especially would like to appreciate detailed observation by Alexander. We might have overlooked the NCA (Growing Neural CA, at Distill.Pub), and acknowledge the similarity between our work (GCA) and that of NCA.
>
> We are currently updating the manuscript with additional experiments, where we surely will reference the work. Before the revision is complete, we would like to summarize the differences between GCA and NCA and reformulate our main contribution here.
>
> 1. Practicality of CA combining neural networks to applications
>
> There have been few works combining neural networks (NN) with CA, like [1] written in the related work section. We think that NCA is one of the first approaches to utilize CA with NNs to reconstruct an image. While intellectually interesting, it is not clear how the combination of CA and NN can be beneficial with NCA. An image can be reconstructed in various ways, such as using conditional GANs or auto-encoders, which has been proven to reconstruct unseen real-world images faithfully, while NCA can recover only a single learnt image.
>
> On the other hand, our work specifically addresses the reasons why we should use the local update rules of CA with NNs. It is inefficient to search the entire 3D voxel space, and GCA exploits the fact that 3D shapes tend to be continuous and sparse. We exploit the connectedness to grow only from the occupied voxel, and utilize sparse CNN [2] to efficiently capture the expressive features. As a result, we obtain (near) SOTA performance in probabilistic shape completion and generation.
>
> 2. Aim of generation
>
> NCA is able to regenerate a single learnt image by updating the state of the neighborhood. Although this approach is promising, their NN only learns patterns of a single image for regeneration and does not generalize to large datasets. However, our method is able to learn to generate / reconstruct multiple high-resolution 3D shapes in a stochastic way. Although a single voxel can have only 2 states (occupied or not), the state space of a shape grows exponentially with the number of voxels.
>
> 3. Perception neighborhood of CA
>
> NCA is a classical CA where each cell only observes its local information. Thus, only local communication between cells is possible. It might be okay for generating a single fixed image, but we need a larger range of perception to build more general CA with diverse modes.
>
> On the other hand, GCA uses a highly expressive U-Net (we will add the full details of the model in the appendix, and plan on releasing the code). As Alexander mentioned, the perception neighborhood is larger than the local update neighborhood (radius r). This enables us to predict a small set of voxels (update neighborhood) that are very likely to be occupied, conditioned on a wide region of shape (perception neighborhood). We think this is a crucial aspect of shape generation. In Appendix D of our paper, we show that GCA is able to generate a globally consistent shape, which would have been impossible with only local information. We realize that the perception neighborhood and update neighborhood has not been fully explained in the manuscript and we will update it.
>
> We will soon upload the manuscript adequately addressing NCA. In addition, we will answer every review and provide a detailed explanation of CA with NN.
>
> [1] N. H. Wulff and J. A. Hertz. Learning cellular automaton dynamics with neural networks. NIPS, 1992.
> [2] B. Graham, M. Engelcke, and L. Maaten. 3d semantic segmentation with submanifold sparse convolutional networks. CVPR, 2018.

---

### Decision · Program_Chairs · 2021-01-07
**Final Decision**

**Decision:**

Accept (Poster)

**Comment:**

This work proposes a method, inspired by Cellular Automata, to generate 3D objects in voxel space. By *only* using local update rule for each location, the method can probabilistic generate high resolution models of everyday objects in the dataset. Due to the ability to incrementally generate details, the quality of the samples are seemingly higher than tradition approaches using Voxel-based GANs.

Most reviewers and myself agree this is a strong and interesting paper that will spark good discussion in the ICLR community. It is also well written and ideas are clearly explained. During the review process, the authors improved the work by conducting additional experiments to analyze the sensitivity of hyper parameters and took in and incorporated various suggestions from the reviewers. After the revision, I believe the work to be in good shape to be accepted at ICLR2021, and I will recommend that this paper be accepted (Poster).